# Towards the Future of Public Health: Roadmapping Trends and Scenarios in the Post-COVID Healthcare Era

**DOI:** 10.3390/healthcare11243118

**Published:** 2023-12-07

**Authors:** Leonardo El-Warrak, Mariano Nunes, Gabriel Luna, Carlos Eduardo Barbosa, Alan Lyra, Matheus Argôlo, Yuri Lima, Herbert Salazar, Jano Moreira de Souza

**Affiliations:** 1Graduate School of Engineering (COPPE), Universidade Federal do Rio de Janeiro (UFRJ), Avenida Horácio Macedo 2030, Centro de Tecnologia, Bloco H, Rio de Janeiro 21941-972, Brazil; leowarrak@cos.ufrj.br (L.E.-W.); gabrielluna@cos.ufrj.br (G.L.); yuriodelima@cos.ufrj.br (Y.L.); herbertsds@cos.ufrj.br (H.S.);; 2Centro de Análises de Sistemas Navais, Rio de Janeiro 20091-000, Brazil

**Keywords:** forecasting, public health, roadmap, future, research avenue

## Abstract

The COVID-19 pandemic, a transformative event in modern society, has disrupted routine, work, behavior, and human relationships. Organizations, amidst the chaos, have innovatively adapted to the evolving situation. However, many countries were unprepared for the magnitude of the challenge, revealing the fragility of health responses due to inadequate leadership, insufficient resources, and poor information system integration. Structural changes in health systems are imperative, particularly in leadership, governance, human resources, financing, information systems, technology, and health service provision. This research utilizes the Technological Roadmapping method to analyze the health sector, focusing on public health, drawing on articles from SCOPUS and PubMed databases, and creating a roadmap extending to 2050. The research presents three long-term scenarios based on the literature-derived roadmap and explores various alternatives, including integrated care, telemedicine, Big Data utilization, nanotechnology, and Big Tech’s AI services. The results underscore the anticipation of post-pandemic public health with high expectations, emphasizing the importance of integrating health history access, encouraging self-care, and leveraging technology for streamlined treatment. Practical implications include insights for decision makers and stakeholders to inform strategic planning and adapt to evolving industry demands, recognizing the significance of preventive services and the humanizing potential of technology.

## 1. Introduction

As far as access to information is concerned, more and more people nowadays may understand the concept of “Health” better. Understanding the behavior of such a valuable asset and the factors that influence it is essential. However, attempting to understand the future of health and the possible impacts it has on the lives of individuals and society is even more challenging.

Public health has made historic advances such as basic sanitation, the discovery of vaccines and medicines, the rapid incorporation of medical technologies for diagnosis and treatment, and the improvement and expansion of health care, among others [1]. Nevertheless, social and demographic factors such as intense urban mobility, faster population growth with increasing life expectancy, high healthcare costs, and growing concerns about mental health challenges, exacerbated by the rapid changes in the world [2], have raised the difficulty of healthcare responses for national healthcare systems. In addition, the advent of new global diseases dramatically pushes public health to be better prepared in the coming decades.

The COVID-19 pandemic has transformed people’s lives as it is an entirely different event in the history of modern society. Similar epidemics have evolved in the past in the context of little integration between countries and people. Changes in routine, work, behavior, and, above all, human relationships were remarkable. Organizations also underwent adaptations, changing processes innovatively amid the chaos caused by the situation [3,4,5]. The pandemic exposed the main weaknesses in health responses in the world [6,7,8]. Even countries with globally recognized public health systems still suffer from low technology incorporation, precarious infrastructure, scarce professionals, and now physically and emotionally overwhelmed ones due to the consequences of pandemics. Because of low investment in public health [9], a large part of the population has not experienced the benefits of advanced technology, and these benefits apply only to the private sector. However, there is an expectation that the pandemic will serve as a lesson for managers and others involved in public health worldwide interested in making the configuration of actions to reduce or mitigate these disparities possible [10,11,12]. Changes in the structuring and organization of health systems have been necessary, especially in leadership and governance, human resources, financing, information systems, products and technology, and the provision of health services.

Decision makers need a thorough and forward-looking understanding of public health. Therefore, the main goal of this work is to forecast the main trends in health care using a literature review in conjunction with the Technological Roadmapping method, and to organize the relevant information as a roadmap. We used the NERMAP online tool (Available online: http://nermap.cos.ufrj.br) [13,14,15] to semi-automate the Technology Roadmapping process by employing Named Entity Recognition [14]. The primary advantage of NERMAP is its capacity to rapidly handle extensive document volumes, leading to substantial time-saving in contrast to manual processing. This enhanced processing speed enables the analysis of content that would be impractical to carry out manually. While there are alternative AI tools, such as Large Language Models (LLMs) like GPT-4 [16], Bard [17], Llama 2 [18], Grok [19] (among others) capable of extracting insights from texts, they come with a significant drawback—cost. Analyzing large volumes of data using LLMs can be prohibitively expensive due to the required computational resources and cloud services. These models often involve substantial operational costs, making them less feasible, especially for projects with budget constraints.

With this work, we hope to make it possible to capture and anticipate future developments in public health to generate insights into how the industry evolves and what options are available to shape the near future.

This work makes significant contributions to the field of public health by addressing crucial aspects of current and future landscapes. Firstly, this work focuses on comprehending the immediate and long-term impact of existing public health challenges on individuals and society, particularly in light of the vulnerabilities brought to light by the recent COVID-19 pandemic. Thus, our research contributes to the broader understanding of how we can reinforce health systems globally. Secondly, the primary goal of forecasting trends in healthcare demonstrates a forward-looking approach that is invaluable for decision makers and stakeholders in the public health sector. The insights generated by this work empower these key players to make informed decisions, plan strategically, and adapt to the industry’s evolving demands. Thirdly, this work’s commitment to organizing pertinent information as a roadmap provides a structured and systematic guide for navigating the intricate landscape of future developments in public health. This contribution facilitates clarity and coherence in the planning and implementation of health strategies. Furthermore, by drawing lessons from the vulnerabilities exposed by the COVID-19 pandemic, this research actively contributes to the ongoing global efforts to better prepare for and respond to future pandemics.

## 2. Materials and Methods

In this work, we analyze scientific papers to gather future trends in public healthcare, applying the Technological Roadmap method to develop post-COVID future scenarios for public health. Technological Roadmapping is a method for identifying, defining, and mapping technological strategies and actions related to innovation [20]. For Phaal et al. [21], roadmapping is a flexible method widely used in management and organizational technology planning. According to Phaal et al. [21], a roadmap is a way to illustrate and communicate technology convergences and timing, guided by a shared vision.

Therefore, we aim to answer the following research questions:RQ1: What are the possible long-term consequences of the COVID-19 pandemic?RQ2: How can we improve global public healthcare to achieve the population’s needs?

To answer these research questions, we based our research methodology, shown in Figure 1, on five steps:Collect relevant literature on public Hhealth by systematically searching scientific databases: SCOPUS and PubMed—science databases relevant to understanding this work’s technological and medical aspects. We removed duplicate documents.Feed the gathered documents into a Technological Roadmapping tool named NERMAP [13,14,15].The NERMAP tool extracts candidate future events and builds an initial roadmap.We analyze and refine the pre-built roadmap, producing a final roadmap.We use the roadmap to develop future scenarios for public health.

In the first step, we aimed to use similar keywords to gather relevant future trends and events related to public health. We chose the SCOPUS database as it comprises various relevant digital libraries across diverse fields of study [22]. Additionally, we included the PubMed database, a biomedical article search engine encompassing over 36 million citations focused on the biomedical literature. By utilizing SCOPUS and PubMed, combined with snowballing procedures, we had a representative set for the proposed search topic [23]. The main keywords used in this research were: “public health”, combined with at least one of the words “future”, “roadmap”, “research avenue”, “research agenda”, or “trend”. We excluded the documents that included the term “review” since they are similar to our research and we are focused on primary studies. We also limited the analysis to papers after 2015 written in English. We considered papers since 2015 because research in healthcare takes longer than in the technological fields. The SCOPUS search returned 209 articles, from which we successfully gathered 157 documents. The PubMed search returned 157 papers, from which we successfully gathered 148 papers. The search strings are shown in Table 1. We joined both results, removing duplicates and inaccessible documents. The inclusion criterion was that the document must discuss the future of public health. The exclusion criteria included documents that were not fully available for download and documents in a language other than English.

**Figure 1 healthcare-11-03118-f001:**
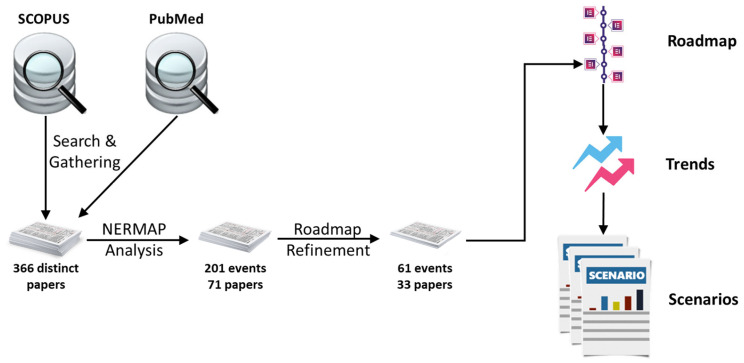
Research methodology.

In the second step, we successfully gathered and inserted 38 unique documents from 2016 to 2022 into the NERMAP tool. The main advantage of using such a tool was the reduced time needed for document analysis. NERMAP has been used in applications such as the technology roadmapping of future Industry 4.0 trends [24,25] and smart and sustainable cities [26]. NERMAP offers a cost-effective solution, an attractive choice for organizations seeking thorough analyses of extensive document volumes that would be impractical to carry out manually. It also leads to substantial time-saving compared to manual processing without compromising financial resources. Additionally, its local functionality ensures data privacy and security, addressing concerns associated with cloud-based alternatives.

In the third step, the NERMAP tool automatically extracted candidate future events using data-referenced phrases from the text. NERMAP automatically selected and placed into the roadmap sentences such as “By 2030, we expect to increase the GDP by 20%”. The NERMAP tool also infered relative dates. For example, the sentence “In the next 10 years, the costs will double”, gathered from a document published in 2017, is considered an event that refers to 2027 in the roadmap.

In the fourth step, we refined the initial roadmap produced by the NERMAP tool. The initial roadmap exhibited two issues common to all information retrieval tools: it could not retrieve some desired results while simultaneously retrieving some undesired ones. Therefore, we used human supervision to merge similar events and remove all misidentified entries as future events. We further manually improved and simplified each event’s description while checking the original document as necessary to provide better contextual information. In this step, we reduced the number of events to 61—from 33 documents.

In the fifth step, the improved events were put into the final roadmap, using the year as the ordering factor. We also further developed the final roadmap into consistent scenarios that used the events as pieces of a complete picture.

We highlight that the roadmap produced by the NERMAP tool relied on the number of documents we uploaded to be analyzed. Adding more documents increased the final roadmap, while removing documents reduced it. However, automation reduced the overhead cost of analyzing more documents, broadening the research reach. Although removing a single document would impact the roadmap, as the number of documents analyzed grows, the impact of the removal of each document would reduce.

## 3. Results

This section presents the roadmap results, which can offer a health map tailored to the public context, particularly in the healthcare sector. We employed the NERMAP tool to generate the roadmap, creating three scenarios based on the perspectives gathered from the recorded events. Broadly, these perspectives aimed to address concerns related to the external environment, disease patterns, political and institutional arrangements of healthcare systems, and population dynamics pertinent to public health on a global scale.

The analysis grew from raw data (roadmap) to more subjective results (scenarios) using the trends as an intermediary step. The development and analysis of individual trends aims to reduce the chance of bias in the analysis. Furthermore, when we built the scenarios, we varied the weight of each trend, producing scenarios with different characteristics. The use of these scenario variations aims to reduce our analysis’s subjectivity and inherent bias. Besides reducing biases, the roadmap of trends and scenarios also systematizes inferences of their consequences, and builds cohesive future settings for public health.

### 3.1. Roadmap

The final roadmap of events is shown in Table 2, covering 2022 and 2050. It is essential to highlight the concentration of events in specific years, even if they come from different citation sources, particularly in 2025 and 2030. The year 2030 is a crucial time marker as it can be influenced by the 2030 Agenda, which consists of a global action plan that brings together 17 sustainable development goals and 169 goals created to eradicate poverty and promote a dignified life. We can also note the trend of citations in years of multiples of five, which can be understood to facilitate the understanding of the triggering of events at regular intervals.

### 3.2. Trends

After building the roadmap, we analyzed the events and extracted 17 main trends from the database of articles that registered some events in the NERMAP. Most of them maintain close relationships, and it is possible to infer that some are consequences of others. Below are the main trends:The population is aging, leading to a rise in the number of individuals with multiple disabilities and long-term care requirements.Mental disorders will increase in incidence and prevalence, particularly depressive disorders and dementia associated with aging, in addition to anxiety conditions, reflecting the effect of the COVID-19 pandemic on the younger population.The incorporation of new technologies will allow the availability of new vaccines.The locus of health action will shifted from medical facilities to households, thanks to the growing popularity of “virtual health”, which includes introducing more home diagnostic tools (self-tests) and telemedicine.There will be a growing need to adopt a more decisive approach toward addressing the causes and risk factors that lead to individual or population health problems. This approach should prioritize primary prevention measures that aim to prevent the development of clinical conditions and early diagnosis and treatment measures for secondary prevention.Information and Communication Technologies (ICTs) will play a leading role in transforming healthcare systems and their care models, enabling the adoption of new technologies that allow for flexible formats of care delivery. This shift will lead to a greater emphasis on value-based care, as opposed to traditional volume-based care, which has helped to improve the overall quality of healthcare services.Regardless of the stage of development of each nation’s search for the best model of care to achieve its ideal health system, the patient will always be considered a central figure.There will be an increase in large tech companies (Big Tech) in the health sector.There will be a significant increase in the utilization of nanotechnology within the healthcare industry.Technological and industrial developments will be made with a significant innovative capacity for public health. Changes in technologies will favor early diagnosis, information and communication technologies, equipment equipped with connectivity, portability, and security, and less invasive equipment.The large-scale use of health data and other data from various diverse, relevant sources will create a multifaceted and highly personalized picture of each user’s health profile and consumption/use of health services.The progression of chronic diseases to more advanced stages, coupled with the failure of healthcare systems to prioritize preventive measures, will increase health costs.Public spending on health will lead to essential changes in the capacity of health systems, especially in the volume and quality of health services.The maintenance of social and economic inequalities between countries. There will be difficulties in establishing more effective sanitary controls. Example: difficulty implementing preventive measures, such as immunization, which are more “homogeneous” between countries.Developing countries will persist in facing endemic diseases, including but not limited to syphilis and other sexually transmitted diseases, malaria, tuberculosis, leptospirosis, leishmaniasis, hospital infections, and arboviruses. New viruses occurring in other countries could result in new pandemics.Governments will broaden their vision beyond their political-administrative limits by strengthening intergovernmental partnerships at the regional level.There will be challenges with complementarity between the public and private sectors, which should ideally function as partners to achieve the best outcomes for universal health coverage. However, finding the balance point for avoiding accentuating social inequalities will still be challenging.

### 3.3. Scenarios

With the events recorded using NERMAP, it is possible to draw some perspectives from identifying the main trends. Therefore, we constructed three alternative scenarios: a possible and optimistic scenario, a likely and neutral scenario, and a plausible and pessimistic scenario.

#### 3.3.1. The Possible and Optimistic Scenario

After two years of advances and setbacks and a lack of a global consensus on drug therapy to treat the infectious disease caused by the SARS-CoV-2 virus, COVID-19, the easy availability of immunobiology, with the spread of vaccination in the largest economies in the world, will enable economic growth to resume. There will be a recovery of global GDP to the pre-health crisis level. Information and communication technologies will generate important reflexes in the health area, especially regarding the quality of service to citizens, efficiency in managing health facilities, and the intelligent use of available information. Therefore, there will be a significant transformation in healthcare based on Big Data, with an accelerated increase in the number and diversity of digitized data that will circulate, be treated, analyzed, and used globally.

The healthcare industry is expected to encounter a surge of data originating from a diverse range of sources, such as electronic medical records, patient activity monitoring via smartphones, real-time alerts, electronic storage of test results and patient data, wearable medical devices, and personalized medicine based on genomic data, among others. New technological possibilities for collecting and analyzing molecular information will expand the options for diagnosing and treating diseases, going beyond the traditional diagnostic tests complementary to treatment. Therefore, the primary locus of health action will be people’s homes, emphasizing remote monitoring and reducing the use of beds and admissions in hospitals. Curative treatments and palliative care will increasingly be replaced by preventive care. Patients, through their profile of consumption/use of health services, will have their therapeutic projects customized, made unique, and adapted to their individual reality. The development of nanotechnology in health will simplify the means of diagnosis and resolve the fight against certain diseases. Genetic mapping will help to increase life expectancy, with years lived being healthier and more affordable for the older population, who will be more active and autonomous. Generally, health levels will improve with the emphasis on health action shifting from medical assistance to health promotion.

#### 3.3.2. The Likely and Neutral Scenario

Population aging will accompany increased chronic conditions (hypertension and diabetes) and mental illnesses (dementia and schizophrenia). However, health systems will adapt slowly to this changing epidemiological profile. Most Latin American and African countries will continue to be unable to universalize vaccination against COVID-19. The triple disease burden (chronic, acute, and external causes) will still challenge these countries. The private sector will concentrate large amounts of patient data. The public sector will still live with the lack of integration of its service network.

Economic and social inequalities between countries will make the staging of technology incorporation in health very different. The dispute over resources (such as vaccines and personal protective equipment) between countries will remain ongoing. Developing countries will not have access to the benefits and consequences of using information and communication technologies in health. The concentration of good practices and innovations will be the domain of countries with the resources to invest in research and technological development. Large technology companies will continue to invest in the healthcare market in developed countries as access to healthcare information becomes more familiar to consumers. Governments will not be able to recover their economies quickly after the pandemic, and the world will become prone to recession and decreased capacity to invest in health.

#### 3.3.3. The Plausible and Pessimistic Scenario

The COVID-19 pandemic will not be under control. All the vaccines will prove inert to new variations of the virus. The extension of social distancing will increase the number of bankruptcies and unemployment, resulting in the gradual loss of membership and giving space to civil disobedience in most countries. The lack of accurate diagnoses for COVID-19 and the emergence of other new infectious diseases, added to the unsatisfactory statistics and lack of integration of databases, will lead to planning failures in the action to fight pandemics, causing the resurgence of the exponential growth of contagion in the world. The mobility of people will be compromised. Due to the volume of calls generated, primary health care will focus on emergencies, with a suspension or limitation of routine consultations and prevention. As a result, there will be an apparent worsening in the world’s population health level, mainly in poorer countries. Rich countries will put up barriers to migration and retain domestic use of supplies to face their health problems. A healthy life increase will not accompany the population aging. The elderly population will considerably pressure social protection and health systems, generating demand for long-term care and high costs.

There will be an excess of disqualified information, generating misinformation in the population. The quality-of-service provision between the private and public sectors will be abysmal. Health will be much closer to suiting the individual than to a collective good. Innovative technologies will only be available to those who can afford them, and inaccessible to many worldwide.

## 4. Discussion

We consider the use of NERMAP for technological forecasting in public health to be quite valuable. The tool could record 61 events in 33 publications. Although NERMAP presents a considerable volume of records, there was a concentration of events in a few articles.

The most frequent events in the roadmap created concern about population aging trends, migration of the locus of health action from health care units to the homes, and the large-scale use of health to create a multifaceted and highly customized sanitary profile and profile of the use of health services for each user. In the following sections, we will briefly discuss the trends most relevant in the search.

### 4.1. Population Aging with an Increase in the Number of People with Multiple Disabilities and Need for Long-Term Care and an Increase in Mental Disorders

Population aging is a worldwide trend due to its predictability, relative inexorability, and the expected impacts of its implementation. An increase in the number of older people brings new health challenges for the public and private health sectors, mainly due to the association between population aging and the rise in demand for specialized and high-quality cost assistance [30,55]. This demand increase is primarily due to the increase, over the years, in the risk of developing chronic diseases or disabilities caused by advanced age [52].

We must prepare healthcare systems to provide good-quality healthcare to older people, with an integration between service providers being closely linked to long-term care sustainability. Integrating the health and social service sectors in a person-centered approach is essential for better care for older people [32,49]. In addition, implementing maintenance-oriented services and improving functional capacity is critical to achieving healthy aging. The older population needs specific care, many of which are specialized and directed to the peculiarities of aging, without segregating those in society [54]. Addressing crucial concerns such as promoting active and independent aging, enhancing the quality of life in older people, balancing the responsibilities of families and governments in caring for the aging population, and recognizing the vital role of older women in caregiving requires a paradigm shift towards perceiving aging as a positive experience. This shift should prioritize continuous health, participation, and safety opportunities to accompany an extended lifespan.

The change in the pattern of diseases, with a more accentuated burden in chronic conditions such as cardiovascular disease and kidney disease, as well as conditions related to mental illness, such as depression, schizophrenia, anxiety disorders, and others, will bring even greater challenges to the health sector [52,53,55]. Aging in the future will take place in a context very different from what we have experienced so far. It will be profoundly affected by climate change or the emergence of new diseases as or more impactful than COVID-19.

It is worth noting that technology can become a great differentiator in the changing of the aging experience. However, barriers to the practicality of its use on a large scale still must be overcome. In anticipation of such changes, health systems and their care models should be equipped to promote healthy lifestyles across all stages of life. These care models include early identification and modification of half-life risk factors to delay disability and provide tailored healthcare and social services that cater to the unique needs of the elderly population. To ensure comprehensive support, integrated and holistic care should be offered across different interaction environments [32,49].

### 4.2. The Migration of the Locus of Health Action from the Health Care Units to the Homes with an Increase in the Use of “Virtual Health” (Development of More Home Diagnostic Tools (Self-Tests) and Telemedicine)

The COVID-19 pandemic had imposed drastic changes and rapid changes in the lives of thousands of people even before reaching the peak of confirmed cases. Hospitals had their structures saturated, unable to absorb the increasing patient demand. For a very long period, the political and health authorities had, as their primary purpose, to stop the contagion spread, mobilizing development centers for immunobiology to make immunizations available in record time for the population [59].

Therefore, home care was essential for vacating beds during the most critical pandemic periods. As hospitals discharged patients with different pathologies who could safely maintain their treatment at home, beds were made available for more critically ill patients who often needed specific equipment such as artificial breathing apparatus. Additionally, the fact that the patients were at home contributed to reducing the risk of contracting COVID-19, as did being in a more controlled and stable environment.

The situation has made more people aware of home care services and benefits. The possibility of receiving medical treatment at home, under the care and company of family members, contributes significantly to the clinical improvement of the patient [61].

The pandemic scenario required companies in the area of health to use technology and innovation in their products and services. There was an acceleration of solutions in health systems that previously seemed impossible to implement in a short time. Jobs in telemedicine and ambulances that travel to the patient’s home for emergency care, allowing the doctor to start the service even before arriving at the patient’s house, are good examples.

Home healthcare is an emerging environment where physicians use handheld and compact devices to treat patients in their homes. Those devices help them check their blood pressure and glucose levels and even perform an electrocardiogram by clicking a button. This technology allows patients to avoid going to the hospital for simpler treatments, while health professionals can go to the patient to administer medication and perform tests. Thus, older people can receive treatment in the comfort of their homes without traveling for hours to be treated [39,44]. Other technological tools, such as applications, chatbots, and new software, were also implemented to facilitate and make the service faster.

The home care sector has a promising future scenario. In some models, provider visits have replaced medical visits, tests, and diagnostics acquired by mobile labs, and imaging and monitoring services of vital signs occur through remote patient involvement [44].

Home care platforms offer at least four promising opportunities for the future of healthcare. Firstly, hospitals will likely limit complex procedures like ICU services and surgeries. Secondly, advancing technology will make it feasible to treat certain diseases and acute conditions at home. Thirdly, telemedicine services are expected to continue growing, enabling remote patient monitoring to expand further. Lastly, there is potential for developing specialized health service segments that utilize patient monitoring data to offer tailored solutions based on individual needs.

### 4.3. The Large-Scale Use of Health Data Together with Others from a Variety of Different Relevant Sources to Create a Multifaceted and Highly Personalized Health Profile and Profile of the Use of Health Services of Each User

The use of Big Data has grown in all areas of science in the last decade, and in health it was no different. Smart devices and sensors help to collect patient data quickly, easily, and effectively [62]. Examining data gathered from these patients is crucial in supporting healthcare professionals, from disease prevention to crafting personalized treatment plans. Numerous simpler electronic devices, like smartwatches and indoor sensors, have Internet connectivity and can be utilized for detecting movement and falls [63]. For older people, for example, if there is a sensor on the house floor connected to the internet, a fall can generate an automatic alert for the caregivers of older people and, through other sensory analyses of this fall, trigger the emergency health service.

The popularization of smartwatches contributes significantly to monitoring a patient’s health, as many of them have accurate heart rate and oxygenation sensors. In this way, such data can be used to accurately measure the patient’s health level, such as their sleep, their heart rate at a specific time, and other vital signs, which may contribute to identifying the risk of heart attacks and vascular accidents.

The amount of data generated could be beneficial for epidemiologists, as it will identify immediate and distant steps that may have led to the emergence of pathologies and even the death of individuals. Furthermore, the use of large amounts of data with the linkage of databases such as that of mortality, notifiable diseases, birth rates, and sanitary and environmental surveillance, among others, will place public health in a privileged position, mainly in matters such as cost–benefit analysis and the impact of health programs on the population. It is clear, then, that using these technologies associated with monitoring and healthcare, brings great benefits, although there are counterpoints. For example, using this information can create difficulties for a select group of people prone to having certain diseases. In this way, health plan operators can benefit from charging abusive prices to specific customers who may be affected by health-related issues in the future [63].

Indeed, the great challenge of using big data in the coming years will be the issue of privacy. There is a real risk of leakage or even theft of confidential data. Perhaps the solution will come with more conservative policies on the subject. Incorporating encryption techniques into database access and searches is recommended. Therefore, caution is needed concerning the use of different data sources, as they can bring benefits or harm to some sectors of society.

On the other hand, when considering the behavioral change trend of the health system user, we need to consider someone who, on their own, often seeks relevant information about their health. These patients want to understand all of their treatment possibilities, risks, and benefits to make better decisions about their health.

With more information, these patients are more likely to recognize the possible risks and problems in their therapeutic projects, thus challenging the maintenance of the traditional health model, where the doctor holds the hegemony of the doctor–patient relationship. In this sense, the patient of the future does not just want to play a passive role: he wants to participate in decisions about his care, and he wants to have access to all information about his treatment.

The traditional health model of curing diseases has not been more attractive to these patients of the future. Patients look for consumer-centric models in which digital resources, tools, and devices are used to create personalized strategies for maintaining health and well-being and guaranteeing attention to their needs, values, and goals. This meaningful change will make service providers have to adapt in a short time.

### 4.4. The Growth in the Use of Nanotechnology in the Healthcare Field

Nanotechnology seeks to manipulate some matters at the molecular or atomic level through nanorobots to solve problems or even create new material from microprocessors. Its main objective would be to master the ability to build systems identical to those in nature. In the healthcare industry, nanotechnology aims to improve human well-being by providing more precise diagnoses, implanting machines for patient treatment, and even developing materials compatible with our biological systems. Biomaterials, made up of nanoscale molecular structures, can interact with biological systems and perform functions like natural mechanisms. These materials can be used in the conformation of various biomedical components, such as blood vessels, artificial skin and organs, smart dressings, vision and hearing devices, and drug delivery systems implanted under the skin [39].

Despite its enormous potential, the use of nanomaterials in health, whether for simple dermatological treatment or for treating diseases such as cancer, is still controversial. The lack of regulation of nanomaterials, the unpredictability of their results, the inherent risk to human and environmental health, and especially the issue of data privacy are still barriers to using technology on a larger scale.

Transformations in society affect a society’s vital needs, caused by changes in living conditions resulting from scientific advances and technological applications such as biotechnology, biomedicine, biogenetics, assisted reproduction, human cloning, and nanotechnology [30,39].

The fact is that nanotechnology has enormous potential to be primarily responsible for developments in health due to its ability to provide benefits to the medical and pharmaceutical areas, improving treatments, diagnoses, surgeries and the use of medicines and materials. For nanotechnology to be applicable in an environment, it requires the use of structures and materials with nanometric dimensions that can penetrate depths previously unattainable by any other biotechnological advancements, significantly improving the quality of life for society.

### 4.5. The Performance of Big Tech in Health

As technology advances on a global scale and more individuals share information on social networks, coupled with the popularity of internet search engines that instantly record web searches and transactions, companies like Google and Amazon use this data to tailor advertisements and search results to each user’s browsing history [64].

According to Arora et al. [64], 80% of US internet users search online for health-related topics, ranging from mental health and immunizations to sexual health information on the Google search engine. In this way, health researchers and policymakers are beginning to see the potential of these data.

This work found a notable trend using the Google Trends (GT) tool associated with public health. The main goal of this tool is to use an online repository of information about user research patterns in real-time, free to use. GT is the primary tool used to study search engine query trends and patterns using Google [65].

GT makes it possible to evaluate the impact of awareness campaigns such as “World Heart Day”, “World Mental Health Day”, “World Diabetes Day”, and “World Hypertension Day”, on the behavior of seeking information on these topics. Therefore, GT offers a high level of geographic accuracy in developed countries, allowing surveys to be stratified at national, regional, and municipal levels. In this way, GT provides a cheaper and faster alternative to data collection to test the effectiveness of public health campaigns [64].

Thus, the GT tool is powerful and can provide valuable information on population health surveillance and behaviors. However, all of the search data available is anonymized and reflects those with internet access, potentially excluding vulnerable groups (e.g., elderly) or regions where internet uptake could be low (e.g., some parts of low- and middle-income countries) [64].

The participation of technology companies in healthcare has been increasing. Thomason [66] reported a 37% growth in the global digital healthcare market in 2021 and projected its value to reach approximately USD 508.8 billion by 2027. There is also the entry of Asia into the digital health market, which already represents 23% of the entire global market, with the forecast that by 2023, it will become the second-largest market and with a more expressive regional growth and an estimated growth of 35% between 2023 and 2028.

The China program (Healthy China 2030) digitizes healthcare to improve efficiency, reduce hospital burden, and encourage relationships between patients, doctors, and the healthy population through wearables, apps, home equipment, and virtual medical schemes. As a result, Chinese technology companies are already building diverse ecosystems for facilitating online medical consultations and managing public health data through advanced AI analytics. In this way, the use of artificial intelligence in manipulating new drugs, creating new technologies for health care, and even monitoring is growing significantly as technology advances.

According to Schuhmacher et al. [67], large companies such as IBM, Microsoft, and Alphabet invest significant amounts in research and development (R&D) and collaborate with pharmaceutical companies in data analysis, diagnosis, management of patients, and health monitoring. Schuhmacher et al. [67] describe some examples of partnerships as The Company Verily—a research organization of Alphabet Inc. dedicated to the studies of life science—which is collaborating with Novartis, Otsuka, and Pfizer on the Baseline project. The Baseline project aims to use technologies to increase patient and physician engagement, shorten study durations, and generate more insightful data to improve clinical research.

Verily is also investing in research with other organizations to discover therapeutics such as bioelectronic medicines to build integrated and intelligent solutions for diabetes management and many other areas of healthcare. IBM uses Watson Health, which provides AI services that assist in diagnostics, personalized treatment, and clinical trials, assisting in developing protocols and clinical trial processes. Another large technology company, Microsoft, has built an ecosystem with several smaller healthcare IT companies to design specific pharmaceutical solutions based on Microsoft’s cloud infrastructure and AI, exemplified by its partnership with Novartis to improve R&D excellence. In conclusion, with this heated market, the trend is for new Big Tech to increasingly enter this field with robust investments and various types of research to facilitate access to digital health.

## 5. Conclusions

In this work, we briefly characterized the health sector, particularly public health, and its possible trends by applying the Technological Roadmapping method, where articles dealing with the topic were inserted and selected in a specific search in the SCOPUS and PubMed databases. It was possible to create a roadmap from the Named Entities Recognition technique. We also presented the results in a timeline, in a horizon that advanced to 2050. A critical analysis was carried out on a series of possible approaches to the assembled roadmap, discussing the efficiency of the NERMAP tool in creating the roadmap and this work’s limitations concerning the events’ sensitivity and the number of databases used.

This work was based on two research questions. To answer RQ1, “What are the possible long-term consequences of the COVID-19 pandemic?”, we developed three long-term scenarios based on the roadmap gathered from the literature. To answer RQ2, “How can we improve global Public Healthcare to achieve the population’s needs?”, we explored the alternatives presented in the Section 4. We covered integrated and holistic care for the elderly and individuals with multiple disabilities, telemedicine/home care platforms, the utilization of Big Data and sensors (such as smartwatches), the enormous potential of nanotechnology, and the involvement of Big Tech’s AI services in healthcare.

This work discusses the future of public health in the world, raising essential aspects that should be better explored, such as the future job market and the increasing incorporation of new technologies. Post-pandemic public health is anticipated with high expectations, and healthcare providers must integrate access to the population’s health history to achieve this goal. Encouraging self-care is also a perspective that poses tremendous challenges since the population’s engagement is essential for the demand for services to be preventive. The new level of citizen awareness and the use of technology proves capable of humanizing care even at a distance, allowing treatment of the population to be streamlined. Finally, the possibility of the emergence of other “pandemics” entails the need to strengthen health systems with an actual increase in the capacity of health actions to track and contain cases and integrate them on a supra-national basis.

This work’s contributions are many. Firstly, this research focuses on comprehending the immediate and long-term impact of existing public health challenges, particularly in light of the vulnerabilities exposed by the recent COVID-19 pandemic. This understanding is pivotal for anticipating and effectively responding to post-pandemic public health expectations. Moreover, this work explores essential aspects such as the future job market and the increasing incorporation of new technologies in public health. This research provides insights for decision makers and stakeholders in the public health sector that can enable decision makers to make informed decisions, plan strategically, and adapt to the industry’s evolving demands. This research also emphasizes the importance of integrating access to the population’s health history and encouraging self-care, recognizing that population engagement is crucial for the success of preventive services. This work also highlights the potential of technology in humanizing care, even at a distance, and streamlining treatment for the population.

Managers in the public health sector should take note of the call for restructuring leadership and governance. This restructuring involves embracing more agile approaches and fostering collaborations to ensure effective responses to evolving challenges. Strategic planning and decision-making processes must consider long-term trends, technological integration, and lessons learned from vulnerabilities exposed during crises. As for practical implications, accelerating technology adoption in public health practices through training programs and infrastructure updates is crucial. Furthermore, fostering collaboration between the public and private sectors is a practical step to ensure that technological advancements benefit all segments of society. Human resources management should focus on supporting and training healthcare professionals, addressing the strain caused by crises, and implementing workload management strategies.

While this research endeavors to provide a comprehensive understanding of the future trends in public health, it is crucial to acknowledge the several limitations inherent in the methodology employed. Firstly, the scope of the literature review may be constrained by access limitations to particular databases or journals, potentially leading to an incomplete synthesis of existing knowledge. Secondly, the reliance on the NERMAP tool for semi-automation introduces a technological limitation, as the tool’s accuracy in identifying and extracting relevant information may be compromised. Thirdly, the potential presence of biases in the selected literature and the data analysis may impact the generalizability of the findings. Finally, the assumption of stability in external factors affecting public health (such as political and social conditions) may not fully capture the dynamic nature of the field, where unforeseen events or emerging challenges could significantly influence the accuracy of forecasts.

This research opens avenues for future research on understanding the long-term impacts of the COVID-19 pandemic on public health, such as assessing the effects on mental health, healthcare infrastructure, and societal health behaviors. Research should explore alternative global health governance models promoting international collaboration and information sharing. Investigating user-centered technology solutions catering to diverse populations is crucial for ensuring equitable access to advancements. Lastly, we evaluated the efficacy of health roadmaps as tools for guiding public health strategies, including their adaptability to unforeseen challenges and their real-world impact on health systems. These research directions can provide valuable insights for shaping the future landscape of public health.

## Figures and Tables

**Table 1 healthcare-11-03118-t001:** Search strings used in SCOPUS and PubMed databases.

Database	Search String
SCOPUS	TITLE (“public health*” AND (“future*” OR “roadmap” OR “research avenue*” OR “research agenda” OR “trend*”) AND NOT “review”) AND (PUBYEAR > 2015) AND (LIMIT-TO (DOCTYPE, “ar”) OR LIMIT-TO (DOCTYPE, “cp”)) AND (LIMIT-TO (LANGUAGE, “English”))
PubMed	(((“public health*”[Title] AND (“future*”[Title] OR “research avenue*”[Title] OR “research agenda”[Title] OR “trend*”[Title])) NOT “review”[Title]) AND 1 January 2015:31 December 3000 [Date - Publication] AND “english”[Language]) AND ((booksdocs[Filter] OR congress[Filter] OR editorial[Filter] OR historicalarticle[Filter] OR letter[Filter] OR news[Filter] OR researchsupportnonusgovt[Filter] OR researchsupportusgovtnon-phs[Filter] OR researchsupportusgovtphs[Filter] OR researchsupportusgovernment[Filter] OR review[Filter]))

**Table 2 healthcare-11-03118-t002:** A roadmap of the events gathered using the NERMAP tool and refined by the authors.

Year	Event	Source
2023	Public expenditure projections reveal that by 2023, 83 out of 189 countries will face contractions in government spending compared to their 2010s average, thereby exposing a cumulative total of 2.3 billion people to the socio-economic consequences of budget cuts.	[27]
2023	Most high-income countries are expected to increase their spending compared to the 2010s. Over half of these countries will have higher government expenditures than pre-COVID averages. Eight countries are projected to experience moderate spending reductions, while the remaining 20 will face aggressive austerity measures.	[27]
2023	Increase in the number of kidney patients affected by the Hepatitis C Virus (HCV) in the US.	[28]
2024	A central objective of Mauritius’s current national health sector strategic plan for 2024 is to review health-financing strategies.	[29]
2025	Increase in the development of genomic medicine, offering a unique opportunity to study the ethical and social issues arising from integrating genomics into routine clinical care in the UK and France.	[30]
2025	As South Africa aims to fully implement National Health Insurance (NHI), and given the pressure on the health sector coupled with limited resources, it will be crucial to define priorities and service packages that address equity.	[31]
2025	In Japan, 30% of the population will be elderly.	[32]
2025	The World Health Organization estimates that approximately 300 million people currently have asthma worldwide, and with current trends rising, it is expected to reach 400 million.	[33]
2025	The TB strategy aims at adopting concrete preventive, diagnostic, and treatment measures to eliminate the disease.	[34]
2025	The surge in government spending due to the pandemic is anticipated to be gradually withdrawn, with countries reverting to their typical expenditure levels as seen in the 2010s.	[27]
2025	Improvments to the optimization process of imaging technologies to increase specificity will be made, thus offering a non-invasive approach to evaluate tumor phenotypes.	[35]
2025	The global prevalence of diabetes is to increase to 570.9 million.	[36]
2025	China will achieve its highest excess growth rate of 2% and increase its GDP% spent on health care from 5.4% in 2012 to 6.6% in 2025.	[37]
2025	The BRICS nations will continue to strive to improve universal and comprehensive health coverage in the upcoming year.	[38]
2025	The transdermal drug delivery (TDD) market is estimated to be worth approximately USD 95.57 billion.	[39]
2027	Many European countries will have rising health inequalities, causing a stall or slightly declining life expectancy. A public health perspective of healthy aging is to remain a priority for European public health in the years to come.	[40]
2027	Racial and ethnic composition and how these affect health and well-being is likely to be one of the critical challenges facing public health leaders in the coming decades in Europe.	[40]
2030	Cancer incidence is expected to increase by aproximately 45% when compared to 2020 numbers, making it the world’s leading cause of death.	[41]
2030	One of the US government’s Healthy People 2030 goals focuses on increasing disease monitoring and prevention efforts, and improving the global capacity to prevent, detect, and respond to public health threats.	[42]
2030	In Mauritius, the 2030 Agenda for Sustainable Development is committed to ensuring universal health coverage, where all individuals and communities, regardless of their circumstances, will receive the health services they need without the risk of suffering financial difficulties.	[29]
2030	Reducing tobacco consumption is essential to reducing noncommunicable diseases in the coming years.	[43]
2030	Health promotion is on the 2030 agenda as a priority goal for sustainable development in Shanghai.	[44]
2030	Results of the projection for 2030 show a significant increase in the costs of treating all “Type 2 diabetes mellitus (T2DM)” and related complications. Accounting for an estimated incidence rate of 55,000 new diabetes cases, the burden of T2DM is projected to cost the South African public healthcare system over USD 2.5 bn by 2030.	[31]
2030	The Sustainable Development Goal is to provide access by 2030 to safe, affordable, and sustainable transport systems for all. An autonomous vehicle policy can seek to address the needs of populations with reduced mobility through targeted subsidies and technical assistance, using transport to make cities inclusive for all.	[45]
2030	A challenge to progress towards “universal health coverage (UHC)” by 2030 is the decrease in OOP spending on pharmaceuticals from the private sector, as there is a general misperception that generic drugs in the public sector are below the norm.	[29]
2030	Indicators such as “noncommunicable diseases (NCDs)”, childhood obesity, the harmful use of alcohol, and mortality from self-harm and interpersonal violence, which suggest that more interventions are still needed to improve patient care in the public health system, are still below the target of 100%.	[38]
2030	SDG targets seek to realize a one-third reduction in premature mortality from NCDs to ensure healthy lives and promote well-being at all ages.	[45]
2030	The target is to reduce global maternal mortality to less than 70 per 100,000 live births, with no country having an MMR above 140 per 100,000 live births.	[46]
2030	It is predicted that 4.5 million South Africans in the public sector will have Type 2 diabetes.	[31]
2030	It is estimated that by 2030 > 70% of patients worldwide with end-stage renal disease will be in developing countries.	[47]
2030	The Association of American Medical Colleges projects a potential national shortage of up to 49,300 primary care physicians and 9600 medicine subspecialty physicians.	[48]
2030	The United States aims to reduce new HIV infections by 75% by 2025 and 90% by 2030.	[49]
2030	Countries seek to double the global energy efficiency improvement rate to mitigate and adapt to climate change.	[45]
2030	Ending the HIV Epidemic.	[49]
2030	Individuals aged 65 and older will constitute 20% of the US population.	[49]
2030	Singaporean seniors aged 65 and above are expected to reach one million.	[50]
2030	The elimination of malaria cases and deaths across Southeast Asia.	[51]
2030	NCDs will be the leading cause of morbidity and mortality, accounting for 80% of deaths worldwide. Cancer affected around 3.45 million Europeans in 2012 and caused 1.75 million deaths. The morbidity and mortality rates of these conditions and other chronic disorders are responsible for the significant burden on public healthcare system expenses.	[51]
2030	About 30,00 seniors will use Long-term Care Facilities (LTCF) or home care services in Singapore.	[50]
2030	An increase in the prevalence of Chronic Kidney Disease (CKD) stages 3a to “end-stage kidney disease” (ESKD) in the United States from 13.2% in 2010 to 16.7% in 2030.	[52]
2030	Tobacco control elements are prioritized in the UN 2030 Sustainable Development Agenda and Sustainable Development Goals.	[43]
2030	The cost of opioid prescriptions for osteoarthritis (OA) is expected to rise to USD 72.4M. The estimated cost per person will grow from USD 62.28 in 2015 to USD 128.68 in 2030.	[53]
2035	For the first time in the history of the United States, people 65 years and older will constitute a greater percentage of the total US population than those 18 years and younger. There will be an expected 78 million people 65 years and older compared with the 76.7 million of those who are younger than 18 years.	[49]
2041	Due to the substantial increase in the Chilean population with chronic kidney disease (CKD), the direct costs of treating CKD are expected to triple.	[52]
2041	The number of people with end-stage renal disease (ESKD), and thus the number of people in need of Renal Replacement Therapy (RRT), is projected to increase from 24,601 in 2021 to 83,885 in 2041.	[52]
2045	The number of individuals with sarcopenia is projected to increase from 19,740,527 in 2016 to 32,338,990. In 2016, the overall prevalence rate among the elderly was 20.2%, and in 2045 it is expected to be 22.3%.	[54]
2045	The proportion of women aged 65 years or older is estimated to increase by 31.4%, from 21.1% in 2016 to 30.0%. The proportion of men in the same age group is projected to increase by 37.0%, from 16.5% in 2016 to 25.2%.	[54]
2050	There are expected to be 1.4 billion Chinese people, with 365 million aged 65+, representing 26.1% of the country’s population.	[55]
2050	The Chinese population statistics taken between 1950 and 2050 show a reduction in the crude death rate and total fertility rate, accompanied by an increase in life expectancy at birth and an expansion of the population aged 65 and above.	[55]
2050	The number of people affected by dementia will reach 135.5 million.	[56]
2050	Dementia diagnoses will rise to 152 million by 2050.	[55]
2050	Without an effective strategy to control antimicrobial resistance (AMR), there will be approximately 10 million deaths each year and a cumulative USD 100 trillion economic burden caused by resistant infections.	[57]
2050	A full implementation of the network, anticipating that digital and mHealth technologies and other smart services will improve prevention, diagnosis, treatment, and management in rural areas in Kazakhstan.	[58]
2050	While the world population has surpassed 7 billion and is expected to continue to increase and reach around 10 billion by 2050, Europe is the only world region expected to experience a population decline by 2050. This decline in population growth is attributed to the low fertility rates experienced in most European countries.	[40]
2050	Nearly 25% of the population in Singapore would have CKD.	[52]
2050	The number of elderly people in India will surpass the population of children below 14 years.	[59]
2050	In India, the proportion of the population aged 60 years and above is projected to increase from 9% to 20%.	[59]
2050	Almost all areas except Africa will have nearly 25% of their people aged 60 or older.	[59]
2050	Over a quarter of the global oldest-old population will live in China.	[55]
2050	Current projections suggest that the population growth rate in African urban regions could swell from 395 million in 2010 to around 1.339 billion.	[60]

## Data Availability

Data are contained within the article.

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
