# Peer review of "Towards the Future of Public Health: Roadmapping Trends and Scenarios in the Post-COVID Healthcare Era"

_healthcare, 2023, doi:10.3390/healthcare11243118_

Round 1
Reviewer 1 Report
Comments and Suggestions for Authors
First of all, I would like to thank you for the possibility of reviewing this interesting paper that I have read with great interest.
The paper may have a clear interest associated to researchers from different scientific disciplines and, therefore, could have a notable repercussion in specialized scientific literature.
Why is this study necessary? should make clear arguments to explain what the originality and value of the proposed model is. This should be stated in the final paragraphs of introduction and conclusion sections.
Literature overview
I would like to suggest the following references:
Fülöp, M. T., Breaz, T. O., He, X., Ionescu, C. A., CordoÅŸ, G. S., & Stanescu, S. G. (2022). The role of universities' sustainability, teachers' wellbeing, and attitudes toward e-learning during COVID-19. Frontiers in Public Health, 10.
Fülöp, M. T., Breaz, T. O., Topor, D. I., Ionescu, C. A., & Dragolea, L. Challenges and Perceptions on e-learning for educational sustainability in a “New Normality Era”. Frontiers in Psychology, 14, 82.
Fülöp, M. T., Udvaros, J., Gubán, Á., & Sándor, Á. (2022). Development of Computational Thinking Using Microcontrollers Integrated into OOP (Object-Oriented Programming). Sustainability, 14(12), 7218.
Method is well
Research limitation I suggest to add it to the end of the research paper
Results and discussion are well
Conclusions: pleas add theoretical, managerial, and practical implications, limitation and further research. Some parts are included but must be extended.
Overall, I believe that the ideas are well expressed, and the storyline is easily followed by the reader. However, in the course of reading the manuscript, I could identify some minor mistakes that should be dealt with more carefully by the authors.
Good luck!
Author Response
Response to Reviewer 1 Comments
Point 1: Why is this study necessary? should make clear arguments to explain what the originality and value of the proposed model is. This should be stated in the final paragraphs of introduction and conclusion sections.
Response 1: We included a paragraph in the introduction and an extra paragraph in the conclusion to argue about the contributions (highlighted in red).
Point 2: I would like to suggest the following references:
Fülöp, M. T., Breaz, T. O., He, X., Ionescu, C. A., CordoÅŸ, G. S., & Stanescu, S. G. (2022). The role of universities' sustainability, teachers' wellbeing, and attitudes toward e-learning during COVID-19. Frontiers in Public Health, 10.
Fülöp, M. T., Breaz, T. O., Topor, D. I., Ionescu, C. A., & Dragolea, L. Challenges and Perceptions on e-learning for educational sustainability in a “New Normality Era”. Frontiers in Psychology, 14, 82.
Fülöp, M. T., Udvaros, J., Gubán, Á., & Sándor, Á. (2022). Development of Computational Thinking Using Microcontrollers Integrated into OOP (Object-Oriented Programming). Sustainability, 14(12), 7218.
Response 2: We included the first two suggested references in the introduction section (highlighted in pink) while broadening the overall references of the section. We included a total of twelve new references in the introduction section.
Point 3: Research limitation I suggest to add it to the end of the research paper.
Response 3: We included a limitations paragraph in the conclusions section (highlighted in green).
Point 4: Please add theoretical, managerial, and practical implications, limitation and further research. Some parts are included but must be extended.
Response 4: We included several modifications in the conclusions section (highlighted in pink, red, cyan, and green).
Reviewer 2 Report
Comments and Suggestions for Authors
Congratulations on the manuscript, it is very interesting, (theme and objectives).
The abstract should state from the beginning the objectives, method, results, conclusions and practical implications. The objectives appear in the last three lines of the abstract.
Much of the abstract should be in the introduction.
The introduction must integrate the research gap, that is, say what the manuscript responds to in terms of the global scientific panorama. Despite sufficient bibliographic references being used, the introduction has only two references.
Refers to the procedure «use NERMAP tool [1] to semi-auto mate the Technology Roadmapping process by employing Named Entity Recognition [2]. The primary advantage of NERMAP is its capacity to rapidly handle extensive document 56 volumes, leading to substantial time savings in contrast to manual processing.», but comparisons to other tools and methodology showold be addressed. As it is, the introduction sens a tutorial for «NERMAP».
Reserach question(s) should be clearer than «we hope to make it possible to capture and anticipate future developments in public health to generate insights into how the industry evolves and the options available to shape the near future».
Methodology has only one bibliographic reference (from 2004). This part should be more populated with references in order not to seem a tutorial for «NERMAP».
Results need systematization (on roadmap and trends). I understand that the «juice» of the research is here, but rationalization of quantity of words and sentences is advisable.
Conclusions are interesting.
Author Response
Response to Reviewer 2 Comments
Point 1: The abstract should state from the beginning the objectives, method, results, conclusions and practical implications. The objectives appear in the last three lines of the abstract.
Response 1: We rewrote the abstract, reducing the context and motivation and including results, conclusions, and practical implications (highlighted in green).
Point 2: Much of the abstract should be in the introduction.
Response 2: We moved part of the "previous version" of the abstract to be included in the introduction section (highlighted in green).
Point 3: The introduction must integrate the research gap, that is, say what the manuscript responds to in terms of the global scientific panorama. Despite sufficient bibliographic references being used, the introduction has only two references.
Response 3: We included twelve citations to support the text in the introduction section (highlighted in pink and cyan).
Point 4: Refers to the procedure «use NERMAP tool [1] to semi-automate the Technology Roadmapping process by employing Named Entity Recognition [2]. The primary advantage of NERMAP is its capacity to rapidly handle extensive document 56 volumes, leading to substantial time savings in contrast to manual processing.», but comparisons to other tools and methodology should be addressed. As it is, the introduction seems a tutorial for «NERMAP».
Response 4: We added a paragraph in the introduction to clarify that we had other alternative AI tools and explained why we chose NERMAP (highlighted in teal). We also reinforced the advantages of using NERMAP in the materials and methods section (also highlighted in teal).
Point 5: Research question(s) should be clearer than «we hope to make it possible to capture and anticipate future developments in public health to generate insights into how the industry evolves and the options available to shape the near future».
Response 5: In the materials and methods section, we divided the research question into two to be clearer (highlighted in pink). In the conclusions section, we summarized the answers to the research questions in the conclusion (also highlighted in pink).
Point 6: Methodology has only one bibliographic reference (from 2004). This part should be more populated with references in order not to seem a tutorial for «NERMAP».
Response 6: We included some references to clarify in which studies the NERMAP tool was applied in sections 1 and 2 (highlighted in dark yellow).
Point 7: Results need systematization (on roadmap and trends). I understand that the «juice» of the research is here, but rationalization of quantity of words and sentences is advisable.
Response 7: We included a paragraph in the results section to clarify how we make our analysis and build our conclusions following a logical flow (highlighted in green).
Reviewer 3 Report
Comments and Suggestions for Authors
In this manuscript, the authors attempt to address the following research question: 'As a consequence of the COVID-19 pandemic, how can global public healthcare be improved to better meet the needs of the population?' To achieve this goal, they initially collected literature from SCOPUS and PubMed. They then utilized a road mapping tool to predict future scenarios.
The methods employed are quite innovative, and the results/scenarios are very impressive. However, I have some concerns:
-
Why didn't the authors use text analysis tools in addition to SCOPUS and PubMed to assess a broader range of literature?
-
Most of the events for the roadmap are derived from individual pieces of literature. I'm curious to know if removing specific literature references, such as reference 5 or 27, would alter the roadmap. It would be valuable for the authors to test the results using a larger dataset, such as 500 or 100 literatures, to assess the differences.
-
I'm also curious about how the authors arrived at their conclusions/scenarios based on the roadmap. Some of the conclusions appear to be derived from the authors' subjective interpretation. It would be beneficial if the authors could describe the detailed prediction process and the results before presenting their conclusions.
-
Lastly, I noticed some minor issues, like double periods. These should be corrected.
Overall, the research is interesting, but addressing these points would enhance the quality and reliability of the study.
Author Response
Response to Reviewer 3 Comments
Point 1: Why didn't the authors use text analysis tools in addition to SCOPUS and PubMed to assess a broader range of literature?
Response 1: We included a text in section 2 that supports the choices of the databases and search methods, providing additional references indicating that the chosen method for analysis is sufficiently representative of the literature search (highlighted in red).
Point 2: Most of the events for the roadmap are derived from individual pieces of literature. I'm curious to know if removing specific literature references, such as reference 5 or 27, would alter the roadmap. It would be valuable for the authors to test the results using a larger dataset, such as 500 or 100 literatures, to assess the differences.
Response 2: We added a paragraph in section 2 to clarify that the impact of removing a single document is reduced as the number of documents analyzed grows (highlighted in cyan).
Point 3: I'm also curious about how the authors arrived at their conclusions/scenarios based on the roadmap. Some of the conclusions appear to be derived from the authors' subjective interpretation. It would be beneficial if the authors could describe the detailed prediction process and the results before presenting their conclusions.
Response 3: We included a paragraph in the results section to clarify how we make our analysis and build our conclusions following a logical flow (highlighted in green).
Point 4: Lastly, I noticed some minor issues, like double periods. These should be corrected.
Response 4: Double periods removed. We also checked the document for grammar issues (highlighted in yellow).